

# A Method for Liquid Spectrophotometric Measurement of Various Forms of Iron and Copper in Ambient Aerosols

Yuhan Yang[1], Dong Gao[2], Rodney J. Weber[1]

[1]Earth and Atmospheric Sciences, Georgia Institute of Technology, Atlanta, 30331, USA

[2]Department of Environmental Health Sciences, School of Public Health, Yale University, New Haven, 06510, USA

*Correspondence to*: Professor Rodney J. Weber, Ph.D. (rodney.weber@eas.gatech.edu)

**Abstract.** Determination of transition metals in ambient aerosols is important due to their toxicity to human health. However, the traditional measurement techniques for metal analysis are often costly and require sophisticated instruments. In this study, we developed and verified relatively low-cost liquid spectrophotometric methods for the measurements of iron (Fe) and copper

(Cu), often the two most abundant transition metals in ambient fine particulate matter ($PM_{2.5}$). For Fe analysis, we utilized a ferrozine based colorimetric method, which has been frequently used for water-soluble (WS) Fe determination,   and further extended this approach for the measurement of total Fe (water-soluble + water-insoluble). In this method, Fe is quantified through the formation of a light-absorbing ferrozine-Fe(II) complex (absorbance at 562 nm). A similar colorimetric method, which forms a bathocuproine-Cu(I) complex absorbing light at 484 nm, was developed and examined for measurement of WS

and total Cu. These methods were applied to 24-hour integrated filter samples collected in urban Atlanta. Based on $PM_{2.5}$ ambient aerosols, total and water-soluble Fe and Cu concentrations were in good agreement with inductively coupled plasma mass spectrometry (ICP-MS) measurements (slopes $1.0 \pm 0.1$, $r^2 > 0.89$). The water-soluble components, operationally defined as those species in the aqueous filter extract that pass through a 0.45 µm pore filter, were further characterized by ultrafiltration, which showed that roughly 85 % of both the Fe and Cu in the water-soluble fraction was composed of components smaller

than nominally 4 nm.

## 1 Introduction

Transition metals are known to contribute to airborne particle toxicity and can cause a wide range of adverse health effects (Chen et al., 2018; Gonet and Maher, 2019).  Iron (Fe) and copper (Cu) are often two of the most abundant transition metals in ambient fine particles ($PM_{2.5}$) and have been linked to a number of human diseases associated with the respiratory,

cardiovascular, and central nervous systems (Calderón-Garcidueñas et al., 2019; Dreher et al., 1997; Wang et al., 2007). They have also been associated with other organ damage (Kim et al., 2013; Schrand et al., 2010). Transition metals in ambient aerosols are emitted in a variety of chemical forms, such as metal oxides, metal sulfates, metal halides, metal organics, and reactive nano-metalloid particles (Gonet and Maher, 2019; Sutto, 2018). Fe and Cu nanoparticles (< 0.1 µm) may be especially toxic due to unique surface properties, combined with their small sizes that allow them to pass through cellular membranes



and ability to be translocated to other organs (Bongaerts et al., 2020; Lee et al., 2020; Moreno-Ríos et al., 2021). Water-soluble species are operationally defined and are obtained though aqueous dissolution of collected aerosols followed by liquid filtration of the extract solution (typical filter pore sizes of 0.22 or 0.45 μm), which is roughly analogous to the process particles undergo when deposited in the respiratory tract lining fluid. Metals measured in this fraction can include molecules and colloidal nanoparticles that are more readily bioavailable than insoluble species. Studies have found that water-soluble transition metals,

including WS Fe and Cu, have stronger adverse health associations than the corresponding water-insoluble components (Costa and Dreher, 1997; Frampton et al., 1999; Huang et al., 2003; Ye et al., 2018). Transition metal ions that can form dissolution of aerosol components, such as Fe(II) and Fe(III) and Cu(I) and Cu(II), are highly redox-active components of $PM_{2.5}$. These ions can trigger and sustain catalytic reaction cycles, such as Fenton reactions, generating reactive oxygen species (ROS; OH, $O_2^-$, $HO_2$, $O_3$, and $H_2O_2$) *in vivo* (Brewer, 2007; Lakey et al., 2016), which can induce oxidative stress and further lead to

inflammatory responses and a subsequent host of adverse health effects (Brook et al., 2010; Donaldson et al., 2001; Ghio et al., 2012; Nel, 2005).

Transition-metal-bearing particles are emitted by a wide variety of sources. Major Fe sources include resuspended mineral and road dust (tire and brake wear), and combustion of biomass and fossil fuels (Khillare and Sarkar, 2012; See et al., 2007; Wang et al., 2016). Cu sources are often dominated by traffic road dust and in certain locations by industrial sources (Adachi and

Tainosho, 2004; Khillare and Sarkar, 2012; Kulshrestha et al., 2009; Lee and Hieu, 2011; Wang et al., 2016). Water-soluble transition metals may originate from primary emissions, such as combustion (e.g., WS Fe), but secondary processing is likely the major source. This can occur by dissolution of the various insoluble forms of the metals emitted as primary emissions, for example, dissolution of metal oxides to soluble forms as the particles age in the atmosphere (Alves et al., 2020; Fang et al., 2015; Galon-Negru et al., 2019; Ito et al., 2019; Meskhidze et al., 2005; See et al., 2007; Wong et al., 2020).

Measuring transition metals in ambient aerosols is challenging due to their low concentrations (10's of ng m$^{-3}$ or less). Various analytical methods have been utilized. X-ray fluorescence (XRF) and particle-induced X-ray emission analysis (PIXE) are non-destructive but can be influenced by matrix effects that result in major interferences (Lin and Luo, 1979). Inductively coupled plasma mass spectrometry (ICP-MS) is widely used for quantifying trace elementals and is included in some federal reference methods for the identification of metals in water and ambient PM in the US (e.g. Method IO-3.5, USEPA Method

1640) (Ammann, 2007; Chow, 1995; Danadurai et al., 2011). Overall, these instruments tend to be expensive, making it challenging to do routine analysis of large numbers of aerosol particle samples in modestly equipped laboratories.

An alternative approach is liquid-based colorimetric methods, which can provide precise and accurate measurements of specific metal species based on the assumption that the ligand used reacts exclusively with the target metal (i.e., in this case Fe or Cu) to form a light-absorbing complex. Stookey (1970) developed a spectrophotometric method for the analysis of Fe(II)

based on Fe(II) complexing with a ferrozine ligand (3-(2-pyridyl)-5,6-bis(4-phenylsulfonic acid)-1,2,4-triazine) that absorbs light at 562 nm. Smith and Wilkins (1953) reported a colorimetric reagent, bathocuproine (2,9-dimethyl-4,7-diphenyl-1,10-phenanthroline), that can selectively chelate with Cu(I). The absorption spectra of the bathocuproine-Cu(I) complex display a maximum value at 484 nm. Both techniques have been developed and used to detect Fe and Cu in natural waters (Moffett et



al., 1985; Viollier et al., 2000), however, use of the Fe-ferrozine approach is much more common and has also been utilized for measurements of ambient aerosols (Kuang et al., 2019; Majestic et al., 2006; Oakes et al., 2010; Rastogi et al., 2009; Zhu et al., 1997; Zhuang et al., 1992). These analytical techniques can be applied to relatively small liquid spectrophotometers that utilize a long path Liquid Waveguide Capillary Cell (LWCC) for high sensitivity. Although only able to measure a specific compound per analysis, in contrast to mass spectrometer methods, the instrument is small, portable, and relatively low cost and can be used for offline measurements of sample extracts or for online measurement when connected to a particle-into-liquid sampler (PILS) or similar system (Oakes et al., 2010; Rastogi et al., 2009). Here we report on the development, verification and application of liquid spectrophotometers for measurement of Fe and Cu in ambient aerosols collected on filters.

## 2 Methods

### 2.1 Collection of Ambient PM$_{2.5}$ filters

One-year of filter collection was conducted throughout 2017 at the Jefferson Street (JST) site located ~ 4 km northwest of downtown Atlanta, Georgia. The JST site is surrounded by a mixed commercial/residential area and has a characteristic urban signature (Hansen et al., 2003). A total of 355 PM$_{2.5}$ filter samples were collected with high-volume (Hi-Vol) samplers (Thermo Anderson, flow rate normally 1.13 m$^3$/min), each filter collected over a 24 h period (midnight to midnight) using prebaked (maximum T = 550 °C) quartz filters (Pallflex® Tissuquartz™, 8×10 inches) with an effective collection area of 516.13 cm$^2$ (20.32 cm by 25.40 cm). Samples were immediately wrapped in prebaked aluminum foil and stored at -18 °C until analysis. A selection of these filters are included in the following analyses. A suite of other air quality sampling instruments was also operational during the filter sampling period. Portions of these filters have already been utilized in studies on aerosol oxidative potential (Gao et al., 2020; Gao et al., 2019).

### 2.2 Reagents and standards

All acids and sodium hydroxide (NaOH) used were trace metal grade obtained from VWR International LLC (Radnor, PA, USA). Standard stock solutions for Fe and Cu calibrations were prepared by dilution of commercially available standards (1000 ppm in 2 % HCl obtained from RCCA chemical company, Arlington, TX, USA) and stored in a refrigerator (T = 4 °C). Calibration with these standards was performed at the beginning of every measurement day. Ammonium acetate (C$_2$H$_7$NO$_2$) was obtained from Fisher Scientific International Inc. (Fair Lawn, New Jersey, USA), and ferrozine (3-(2-pyridyl)-5,6-bis (4-phenylsulfonic acid)-1,2,4-triazine) from the HACH company (Loveland, CO, USA). Bathocuproine (2,9-dimethyl-4,7-diphenyl-1,10-phenanthroline), sodium citrate dihydrate (HOC(COONa)(CH$_2$COONa)$_2$·2H$_2$O) and hydroxylamine hydrochloride (HONH$_2$·HCl) were obtained from Sigma Aldrich (St. Louis, MO, USA). Details of the chemical preparations are provided in the Supplement.


## 2.3 Spectrophotometer and LWCC system

Similar spectrophotometric instruments were used for all measurements (WS and total Fe and Cu). The final mixture containing
the light-absorbing complexes were pushed into a Liquid Waveguide Capillary Cell (LWCC-3250/ LWCC-3100, World
Precision Instrument, Sarasota, FL). The waveguide was coupled at one end to a dual deuterium and tungsten halogen light
source (DT-Mini-2, Ocean Optics, Dunedin, FL), which produced light over the wavelength range of 200 to 800 nm, and at
the other end to a multi-wave light detector (USB4000, Ocean Optics, Dunedin, FL), using 400 µm fiber core-diameter fiber
optic cables (QP400-2-SR, Ocean Optics, Dunedin, FL). The light absorbance was recorded with a data requisition software
(SpectraSuite). For Fe analysis, the light absorption over the wavelength range between 557 nm and 567 nm was measured
and averaged as the absorbance of the Fe complex in the sample ($Abs_{562}$). Average absorption between wavelengths of 479
nm to 489 nm was recorded and referenced to as the absorbance for Cu measurements ($Abs_{484}$). In both measurements, the
average absorbance measured over the wavelength range of 695 nm to 705 nm ($Abs_{700}$) was chosen as the baseline absorbance.
Since the light absorption efficiencies of Fe(II)-ferrozine and Cu(I)-bathocuproine complexes differed, the optical path lengths
of the LWCC for Fe and Cu measurements were 100 cm (LWCC-3100) and 250 cm (LWCC-3250), respectively. To keep the
light absorption within an optimal range, the recommended Fe(II) and Cu(I) liquid concentrations detected by the system are
up to 40 and 50 ppb, respectively.

The metal concentrations in the liquid ($C$) were determined based on the light absorbances by:

$$C = \frac{1}{b}[(Abs_\lambda - Abs_{700}) - a]. \quad (1)$$

Where $Abs_\lambda$ represents the average light absorption of the sample (or blank) for the ranges given above, which encompasses
the wavelength ($\lambda$) at which maximum optical absorption occurs; for Fe, $\lambda = 562\ nm$, and for Cu, $\lambda = 484\ nm$. $Abs_{700}$
represents the average light absorption between 695 to 705 nm, $a$ (arbitrary units) and $b$ (1/ng mL$^{-1}$) are the intercept and slope
of the calibration curve, respectively.

The final ambient concentration of each element, $C_{air}$ (ng m$^{-3}$), was calculated by:

$$C_{air} = (C_{sample} - C_{blank})\frac{V_l}{V_s} \times \frac{V_e}{V_{air}} \times \frac{A_{filter}}{A_{punch}}. \quad (2)$$

Where $C_{sample}$ and $C_{blank}$ are the liquid sample and blank concentrations of the specific element (ng mL$^{-1}$), determined by
Eq. (1), $V_e$ (mL) is the liquid volume of the sample (or blank) extract, $V_s$ (mL) is the liquid extract volume used for the actual
measurement, and $V_l$ (mL) is the volume of liquid sample solution after being diluted by all the chemicals (i.e. $V_l = V_s +$
$V_{chemicals}$). $V_{air}$ (m$^3$) is the volume of air drawn through the filter during sampling. $A_{filter}$ (cm$^2$) and $A_{punch}$ (cm$^2$) are the
total areas of the Hi-Vol filter and the filter punches used for analysis, respectively.



### 2.4 Filter extraction

### 2.4.1 Extraction for water-soluble metals and ultrafiltration

For measurements of the water-soluble Fe and Cu collected on the filters, one circular filter punch (1 inch in diameter, $A_{punch} = 5 \ cm^2$) was extracted in 7 mL ($V_e$) of de-ionized water (DI water, Nanopure Infinity™ ultrapure water system; >18

MΩ cm$^{-1}$) in a sterile polypropylene centrifuge tube (VWR International LLC, Suwanee, GA, USA) via 30-min sonication (Ultrasonic Cleanser, VWR International LLC, West Chester, PA, USA). The extract was filtered using a 0.45 µm PTFE syringe filter (Fisherbrand™) and then acidified using 6 M high purity HCl to a pH of 1 to preserve the Fe or Cu in solution. A schematic of the extraction is shown in Figure 1.

The filterable metal fraction in the extracts, defined as water-soluble metals in this paper, will include all dissolved metal forms

and any colloidal particles with a diameter less than 0.45 µm, assuming that all colloidal particles < 0.45 µm can penetrate through the syringe filter and the retention efficiency of particles > 0.45 µm is 100 %.

To assess the contribution of colloidal particles to the operationally defined WS Fe and WS Cu, a subset of the collected filters (N = 69) were further filtered by ultrafiltration (Amicon™ ultra centrifugal filter units, Merck Millipore Ltd., Tullagreen, Carrigtwohill, Co. Cork, IRL). Two filter sizes, 30,000 Daltons (Da) and 3,000 Da, were used for size fractionation, which

roughly correspond to colloidal particles of 4 nm and 2 nm diameter, respectively (Erickson, 2009). Details of the conversion of particle mass in Da to particle diameter and the effects of ultrafiltration membrane rejection are discussed in the Supplement. The particles were separated within the 30 kDa and 3 kDa ultrafiltration units at ~ 3000g (4000 rpm) centrifugation for 30-min and 60-min, respectively. The filtrate was then pH adjusted to 1 and measured for WS Fe and WS Cu.

The soluble metal fraction may be affected by the degree of dilution of the samples in the extraction process. The ratio of

extraction water volume to the volume of air sampled on the analyzed filter portion, denoted as $P$, is ~ 0.5 in this study. This implies that Fe- and Cu-containing substances with solubility higher than roughly $10^{-3}$ g L$^{-1}$ under conditions close to neutral pH will likely be fully dissolved, while species with water solubility as low as $10^{-6}$ g L$^{-1}$ may be partially soluble but still contribute to the WS Fe and WS Cu measured with our method. Details of the effect of $P$ on Fe or Cu solubility are discussed in the Supplement.

### 2.4.2 Extraction for total metal analysis

The extraction procedures for the determination of total Fe and Cu (shown in Fig. 1) follow the method conducted by Aller et al. (1986). To increase the solubility of metals, filter punches (1.5 cm$^2$ for Fe and 3 cm$^2$ for Cu) were digested with 1.5 mL and 2 mL of 6 M HCl for total Fe and Cu measurements, respectively. The filter was incubated in acid at 99°C and shaken at a rotational frequency of 350 rpm using a ThermoMixer (Eppendorf North America, Inc., Hauppauge, NY, USA) for 24 h.

This extraction method has been demonstrated to remove >96 % of Fe from the PACS1 international sedimental standards and > 98 % of Fe from standard rocks and inner shelf muds (Poulton and Canfield, 2005). The effect of extraction time, heating temperature and rotation rate on extraction efficiency were tested and are discussed below. The extract was diluted to 10 mL



$(V_e)$ using DI and then filtered by a 0.45 μm pore PTFE syringe filter. The volume of the sample extract used for the measurement $(V_s)$ differed between samples and was determined by the ambient concentration of each sample. The pH of sample extract was adjusted to 1 using 5 M NaOH, and the sample volume was adjusted to 2.5 mL with DI water.

A total of 10 blank filters were collected throughout the one year of sampling, and these were used randomly to assess the blank levels (i.e., multiple punches were taken from the blank filters for a blank analysis on different days). The same tools and type of filter were used for blanks and samples to track all possible contamination. The concentration of blanks was calculated using Eq. (1) assuming $V_a$ was equal to the volume of air drawn through actual ambient filter samples.

## 2.5 Colorimetric Methods

### 2.5.1 Colorimetric Method for Fe

A ferrozine based method, which has been previously established and discussed (Oakes et al., 2010; Rastogi et al., 2009), was adopted and modified in this study for Fe measurement. Briefly, excessive amount of reducing agent, hydroxylamine hydrochloride (0.1 mL; 1.5 M), was added to 2.5 mL $(V_s)$ of sample solution to reduce all soluble forms of Fe (e.g., Fe(III)) to Fe(II). The vial was sealed and stored overnight (roughly 18 hours) at room temperature to ensure complete reduction (Bengtsson et al., 2002; Rozan et al., 2002). After the overnight redox reaction, the sample pH was adjusted to 4 ~ 5 using ammonium acetate buffer. 0.2 mL of 5 mM ferrozine-acetate reagent was added to the solution to form the colored ferrozine-Fe(II) complex that absorbs light at 562 nm. The total volume of sample solution after addition of all the chemicals $(V_l)$ was 2.85 mL.

The reaction between Fe(II) and ferrozine is found to be pH-dependent; the absorption of the complex increases rapidly when pH is above 0 and becomes steady within the pH range of 3 to 6 (Gibbs, 1976; Stookey, 1970). Under the conditions of the original method, when the pH of the sample was approximately 1, the absorbance of the complex was sensitive to small fluctuations in pH, adding uncertainty to the analysis. A further disadvantage is that at lower pH (pH = 1), the reaction rate of Fe(II) and ferrozine was much slower relative to higher pH (3-5). Therefore, we tested a modification to the method by raising the pH of the sample to 4 ~ 5 during the colorimetric reaction using ammonium acetate buffer to increase the precision of the Fe measurement (Fig. 1). Part of the WS Fe samples was analysed both ways, with low and higher pH, whereas all total Fe measurements were done with the buffered (higher pH) samples.

### 2.5.2 Colorimetric Method for Cu

For the determination of Cu concentrations, a bathocuproine assay was used. Bathocuproine can selectively chelate with Cu(I) and form complexes absorbing light at 464 nm. To convert other soluble Cu forms (e.g., Cu(II)) to Cu(I), 0.5 mL of 1.6 M hydroxylamine hydrochloride was added to 2.5 mL $(V_a)$ sample extracts at pH 1 (as described above) and allowed to react for 30 minutes (Imamura and Fujimoto, 1975; Tomat and Rigo, 1975). The reaction between Cu(I) and bathocuproine is also pH-



dependent, and so a set of controlled trials were performed with different pH conditions ranging from 0.5 to 6 to assess the optimal pH for forming the colored complex (Fig. S1). The light absorption of the complex remained low at pH < 1.5 and

exhibited a rapid increase above pH 2 and became steady when pH was within the range of 5 to 6. Based on these results, 1 mL of 1 M sodium citrate was added to the sample solution to maintain the pH within the range of 5 to 6, followed by 1 mL of 2.8 mM bathocuproine ($V_l = 5.05\ mL$). After 15-min reaction, the light absorption of the bathocuproine-Cu(I) complex was measured.

The colorimetric methods can provide insights on the forms of metal ions present, such as Fe(II) and Fe(III), or Cu(I) and

Cu(II). For example by first measuring the reduced species, Fe(II) or Cu(I), then adding the reducing agent to measure the total (Fe(II)+Fe(III), or Cu(I)+Cu(II)). We do not attempt to speciate the WS Fe and WS Cu in this study since reduction-oxidation processes during the sampling, storage or extraction procedures may alter the valence states of airborne metals from what existed in the ambient particles.

**2.6 Metal analysis by ICP-MS**

Both total and water-soluble elements, which included magnesium (Mg), aluminum (Al), potassium (K), manganese (Mn), and zinc (Zn) in addition to Fe and Cu, were also measured by inductively coupled plasma mass spectrometry (ICP-MS) (Agilent 7500a series, Agilent Technologies, Inc., CA, USA). The sample preparation procedures have been described in detail in Gao et al. (2019). The sample preparation for WS metal analysis followed the same extraction procedures as those described above for WS Fe and Cu. The filtered extract was then acid-preserved with concentrated nitric acid (70 %) to a final

concentration of 2 % (v/v). For total metal analysis, aqua regia, a more powerful acid than HCl, was used for filter digestion. The acid-digested sample was then diluted and filtered for ICP-MS analysis.

**3 Results and Discussions**

**3.1 Evaluation of the colorimetric measurement method**

**3.1.1 Calibration of the Detection Methods**

The analytical systems were calibrated and assessed with a series of standards diluted from the standard stocks of Cu(II) and Fe(III). The calibration curves for the reduced form of metals (Cu(I) and Fe(II)) are illustrated in Fig. 2. York regression (Wu and Yu, 2018) was applied and demonstrated a strong linear relationship between the element concentrations and the detected light absorbance with the coefficient of determination ($R^2$) of 1.00. The slopes of calibration curves from multiple calibration runs were $0.033 \pm 0.004$ for Cu (coefficient of variation, CV = 12 %, N = 22) and $0.053 \pm 0.005$ for Fe (CV = 9 %, N = 5),

indicating the stable performance of the systems. Repeated measurements (N = 20) of a 35 µg L$^{-1}$ standard solution had a relative standard deviation for both Cu and Fe of < 10 %. These results indicated good precision and the robustness of this method for quantifying both Cu and Fe.



### 3.1.2 Blanks, Detection Limit and Uncertainty

Field blanks were analyzed on each test day in parallel with samples to assess the overall system background. The limits of
detection (LODs) of the spectroscopic systems, determined as three times the standard deviation of the filter blanks, were 3.07
ng m$^{-3}$ (N = 34) for WS Fe, 14.79 ng m$^{-3}$ (N = 6) for total Fe, 3.04 ng m$^{-3}$ (N = 29) for WS Cu, and 3.23 ng m$^{-3}$ (N = 6) for
total Cu. The overall uncertainties of the sampling system ranged from 7 % to 14 %, calculated from the combination of the
analytical uncertainty, uncertainty of blank measurements and quadrature sum of square of various relative uncertainties (such
as uncertainties caused by calibrations, blank filters, flow rates, liquid extract volumes, etc.). The detection limits and
uncertainties were comparable to the measurement using ICP-MS, indicating that LWCC has an equally good performance as
ICP-MS, especially for the WS Fe measurement. A summary of LOD, blank and uncertainty are shown in Table 1.

### 3.1.3 Interference

In the spectrophotometric method, species in ambient aerosol particles, such as black carbon (BC, or also called refractory
BC), light-absorbing organic species (i.e., Brown Carbon, BrC) or inorganic species (i.e., components of mineral dust), can
also absorb light at 482 or 562 nm, potentially resulting in measurement interference. However, these interferences are not
accounted for in the blanks since no particles are collected as part of the blank measurement. To assess the interference, the
light absorption at 484 or 562 nm of the individual liquid samples was measured without adding the colorimetric reagents (see
X in Fig 1). The interference was measured for the ambient samples collected in this study and in an extreme case (sampling
wildfire smoke). By comparing the interference corrected and non-corrected data, it is found that the average interference was
less than 20 % for WS Fe and 1 % for total Fe. For total and WS Cu, a correction of about 24 % and 30 % was found,
respectively. For the extreme case, samples from within wildfire plumes, for WS Cu an average correction of 50 % was needed
(see supplemental Fig. S2). The impact of this correction also depends on the concentrations of the Fe or Cu, which in smoke
plumes can be at low concentrations relative to high level of light-absorbing aerosol species (i.e., BC and BrC) and so the
correction, in this case, is important. The magnitude of the interference also depends on the extraction method; methods that
more effectively extract solid particles (e.g., BC) or solubilize light-absorbing species at low pH (e.g., mineral dust) could
produce higher interferences. Black carbon absorbs light over all wavelengths, whereas BrC and mineral dust absorb
preferentially at lower wavelengths and so would be a potentially larger interference for the Cu method since the Cu complex
is measured at 484 nm compared to 562 nm for Fe. Since incomplete combustion is the main source for BC and BrC, tests for
this interference are most important when these types of emissions make a large contribution to the overall aerosol sources,
such as sampling in regions highly influenced by biomass burning or vehicle tailpipe emissions. Interference from light
absorbing mineral dust species is most likely when sampling under dry conditions and high winds.
We also used a series of metal mixtures, which contained different levels of Fe and Cu, to test the possible interferences from
the formation of colored complexes with other metals. For the Cu measurement, no interference from Fe was observed with
up to 100 ppb of Fe present. Similarly, Cu did not interfere with the Fe measurement with Cu concentrations in the liquid up





to 100 ppb. In this study, the liquid concentrations of Cu and Fe from ambient samples were much lower than 100 ppb, therefore, no interference between Cu and Fe for the colorimetric methods. All measurements for ambient samples in this study were blank and interference corrected.

### 3.1.4 Comparison to ICP-MS Measurements and influence of experimental conditions

To further evaluate the accuracy of the new detection method, the concentrations determined by the colorimetric methods were
compared to ICP-MS measurements. WS Fe and WS Cu were measured with the colorimetric methods using the LWCC system on all collected filters (N = 355), but only a portion of the 355 filters was used for the measurement of total Fe and total Cu using the same LWCC system (Fe: N = 23, Cu: N = 21). All filters (N = 355) were used for the ICP-MS measurements of both WS and total elements.

*Water-soluble Fe and Cu:* Regression analysis between the colorimetric methods and ICP-MS measurements showed good quantitative agreement for WS Fe and WS Cu with $R^2$ greater than 0.89. The differences between the LWCC and ICP-MS measured concentrations were 6 %, and 12 % for WS Fe and WS Cu (Fig. 3a and 3c) with the offsets (intercepts) less than 1 ng m$^{-3}$. Considering measurement uncertainty, most points overlap the 1:1 line except for several WS Fe measurements. The difference in WS Fe between these two methods was possibly due to the acidic conditions (pH ~1) of the colored complex
forming solution used in the original approach of the colorimetric method. As noted above, to improve the method, some of the samples (N = 69) was re-analyzed with sample pH adjusted to 4 ~ 5, which resulted in a higher correlation to the ICP-MS measurement, consistent with less sensitivity of the complex to pH when in the higher pH range ($r^2 = 0.92$, slope 1.02, intercept -0.42 ng m$^{-3}$) (Fig. 4). Considering the overall uncertainties of these two methods, the LWCC and ICP-MS measurements of the water-soluble metals were in agreement.

*Total Fe and Cu:* Similar results were found between the LWCC and ICP-MS methods for the measurement of total element concentration. For total Fe, York regression between LWCC and ICP-MS yields a slope of 1.07, with an intercept of 3.09 ng m$^{-3}$ and $r^2 = 0.97$, see Fig. 3b. For Cu, the slope was 0.96, intercept = -1.86 ng m$^{-3}$ and $r^2 = 0.95$ (Fig. 3d).

As noted, in the colorimetric methods description above, the filters for the total Fe and total Cu analysis were extracted for a 24 h period with the assistance of shaking and high temperature. The effect of extraction time, heating temperature and rotation
rate on extraction efficiency were tested (with N = 3 filters for each condition). It was found that for total Fe extraction, there was a 5 % decrease when the extraction time was shortened from 24 to 2.5 hours and a 5 % increase when extraction time was increased from 24 to 120 hours. For Cu, the concentration decreased by 15 % when the extraction time was changed from 24 to 2.5 hours and decreased by 5 % when the extraction time was extended to 120 hours. An extraction time of 24 hours was concluded to be optimal for both Fe and Cu. Extract temperature had a small impact, with a 5 % and 10 % decrease for Fe and
Cu extraction, respectively, when the temperature decreased from 99℃ to 50℃. The rotational shaking was found to significantly affect the extraction efficiency. Compared to shaking at a rate of 350 rpm, extraction without rotational shaking reduced the extraction efficiency by 98 % and 89 % for Fe and Cu, respectively.





### 3.2 Characteristics of Fe and Cu in Urban Atlanta

#### 3.2.1 Water-soluble, Total and Fractional Solubility

The annual average concentrations of total Fe measured by ICP-MS and WS Fe measured by LWCC was 203.7 ng m$^{-3}$ and 20.2 ng m$^{-3}$, respectively. Mass concentrations of WS Fe spanned a wide range, from the detection limit (3.07 ng m$^{-3}$) to 169 ng m$^{-3}$. The average concentrations of total Cu and WS Cu were 30.7 ng m$^{-3}$ and 13.8 ng m$^{-3}$, respectively. Mass concentration of WS Cu ranged from the detection limit (3.04 ng m$^{-3}$) to 132 ng m$^{-3}$. Table 2 and Table S1 provide a more complete summary. The statistics of the LWCC measurement and ICP-MS measurement of total metals were different since only a fraction of all

the 2017 filter samples were analyzed for total metals using the LWCC system (Fe: N = 23, Cu: N = 21). These concentrations were comparable with a previous study conducted in Atlanta, GA, and in the range reported for many other regions globally (Table 3).

As expected the total Fe concentration was much higher than total Cu since Fe is a common element in the Earth's crust (mass fraction of 5.63 %) (Taylor, 1964) and ubiquitous in mineral dust, while PM$_{2.5}$ Cu in urban environments is mainly derived

from vehicle brake and tire wear, and motor-oil impurities resulting in tail-pipe emissions; both of these sources are associated with traffic (Fang et al., 2015; Marcazzan et al., 2003). Other industrial sources of PM$_{2.5}$ Cu are also possible. Despite much higher total concentrations, the concentrations of WS Fe were comparable with WS Cu. This is due to a much lower solubility of Fe than Cu. The average soluble fraction of Fe and Cu are 14.6 % and 50.7 %, respectively (Table 2).

#### 3.2.2 Components of Water-Soluble Fe and Cu Based on Ultrafiltration

Ultrafiltration of WS Fe and WS Cu was used to assess contributions of colloidal particles to the concentrations of water-soluble metals. The size fractionation results are given in Fig. 5. Based on medians of all ultrafiltration data, approximately 82 % of the WS Fe had a nominal diameter smaller than 4 nm, passing through the 30k Dalton filter; 56 % of the WS Fe were nominally smaller than 2 nm in diameter, which passed through the 3k Dalton filter. The fractions for the sizes in between were calculated by taking the difference, and a rough distribution of Fe in the overall WS Fe was: 56 % smaller than 2 nm, 26

% between 2 and 4 nm, and 18 % between 4 nm and 0.45 μm. For WS Cu, most (82 %) of WS Cu passed through the 3k Dalton filter and had a nominal diameter less than 2 nm, with 2 % between 2 and 4 nm and 16 % between 4 nm and 0.45 μm. The 2 to 4 nm fraction of Fe colloidal particles (23 %) was higher than that found in WS Cu (4 %, which was practically zero). Laboratory studies find that ambient particles undergoing pH cycling from low pH (~ 2) in deliquesced particles to circumneutral pH of ~5-6 when activated in cloud drops leads to the formation of nanoparticles of highly reactive ferrihydrite

of a few nm diameter (~5 nm) (Shi et al., 2011). The pH of the bulk PM$_{2.5}$ aerosol in this region is estimated to be in the range 1.63 ~ 2.31 (25th and 75th percentiles throughout the year (Wong et al., 2020)), which would rise rapidly to pH between 5 and 7 in the extraction solution (based on measurements). Rapid increase to circumneutral pH leads to a supersaturated soluble Fe(III) which then precipitates out of the solution. Thus, either from pH cycling experienced by the ambient aerosol, or changes due to the filter extraction process, the colloidal particles of WS Fe observed in the 2 to 4 nm range are consistent with the



ferrihydrite reported by Shi et al. (2011). Acid driven solubility of Cu occurs rapidly at higher pH's and so may not reach supersaturation like that of Fe when pH increases to circumneutral, accounting for the lack of colloidal particles in the nominal 2 to 4 nm size range WS Cu.

For both Fe and Cu, the fraction of Fe and Cu in the 4 nm to 0.45 µm range may be insoluble chemical forms from primary emissions, like metal oxides that had been extracted from the filter and had passed through the syringe filter, making them not

a "true" water-soluble species (i.e., not unique from the insoluble fraction). More research needs to be done to determine what caused the variability in the various fractions separated by ultrafiltration and to characterize the forms of the metals in the various size fractions of what we operationally define as overall water-soluble metal. These results have implications for aerosol toxicity and health effects.

## 4 Conclusion

In this study, a new approach for the measurement of WS Cu and total Cu has been developed and tested based on liquid spectrophotometry. It follows a similar method for quantifying WS Fe. All measurements can be performed with a single relatively inexpensive spectrophotometer. Utilizing daily $PM_{2.5}$ filters collected in Atlanta during 2017, the spectrophotometer methods were assessed by comparing with elements measured by ICP-MS on the same ambient filter samples. Calibrations with liquid standards produced by serial dilutions were highly linear and stable over periods of days. The spectrophotometric

method detection limits in this study were 3.07 ng m$^{-3}$ for WS Fe, and 14.8 ng m$^{-3}$ for total Fe, 3.04 ng m$^{-3}$ for WS Cu, and 3.23 ng m$^{-3}$ for total Cu with measurement uncertainties ranging from 7 to 14 %. The detection limit and uncertainties are comparable to ICP-MS measurement. Possible interferences from other light-absorbing species, such as black and brown carbon should be considered, especially for the Cu measurement and in situations where emissions of light-absorbing aerosols are high (e.g., BC, BrC, mineral dust). From comparisons with ICP-MS measurements, in all cases (WS Fe, total Fe, WS Cu

and total Cu) the spectrophotometry data were in good agreement (slopes ranged between 0.88 and 1.07 and r$^2$ > 0.89).

In Atlanta, total Fe concentration was much higher than total Cu, whereas the level of WS Fe was comparable with WS Cu, due to a higher soluble fraction of $PM_{2.5}$ Cu (45.6 %) compared to Fe (14.6 %). The composition of what is operationally defined as WS Fe and WS Cu (i.e., the aerosol filter extracted in pure water and extract passed through 0.45µm pore liquid filter) was investigated with ultrafiltration Fe and Cu in the filtrate measured with the spectrophotometric (LWCC) method.

Ultrafiltration showed that roughly 82 % of WS Fe and 84 % of WS Cu was composed of dissolved species or colloidal particles smaller than nominally 4 nm diameter. Fe and Cu are two important redox-active metals that have been linked to reactive oxygen species (ROS) formed *in vivo* and lead to adverse health effects through oxidative stress. These spectrophotometric methods allow analyses of the various physical forms of Fe and Cu in ambient aerosol particles and are directly applicable to aerosol toxicity studies and their contribution to aerosol oxidative potential (OP).



## Data availability

The data used in this paper are publicly available at https://github.com/yhyang611/A-Method-for-Liquid-Spectrophotometric-Measurement-of-Various-Forms-of-Iron-and-Copper-in-Ambient-Ae/blob/main/summaryofdata(1).xlsx and can also be obtained from the corresponding author upon request.

## Author contribution

YY is responsible for the study design, experiment using LWCC, data analysis and manuscript writing. RJW supervised the study and participated in the writing and revision of the paper. DG provided ICP-MS data and interpretation of the results.

## Competing interests

The authors declare that they have no conflict of interest.

## Acknowledgements

This work was supported by the NSF under grant number 1927778 and by the Georgia Tech EAS Jefferson Street Funds through a generous gift from Georgia Power-Southern Company.

Thanks are given to Linghan Zeng for his kind guidance in experiment and many interesting discussions about data analysis. We would also like to thank Allison Weber for her help in conducting the IC measurements.

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





**Table 1:** Limits of detection, blanks and uncertainties for WS Fe, WS Cu and total Cu for the liquid spectrophotometric (LWCC) and ICP-MS measurement

|  | LWCC | | ICP-MS | |
|---|---|---|---|---|
|  | LOD, ng m$^{-3}$ | Overall uncertainty, % | LOD, ng m$^{-3}$ | Overall uncertainty, % |
| **WS Fe** | 3.07 | 12.5 | 4.38 | 18.0 |
| **WS Cu** | 3.04 | 13.8 | 2.86 | 6.3 |
| **Total Fe** | 14.79 | 7.2 | 14.80 | 7.2 |
| **Total Cu** | 3.23 | 9.9 | 9.67 | 6.3 |

**Table 2:** Statistical summary of transition metal concentrations in PM$_{2.5}$ measured in Atlanta in 2017 based on liquid spectrophotometric (LWCC) measurement.

|  | Total Fe (N = 23) | Total Cu (N = 21) | WS Fe (N = 355) | WS Cu (N = 355) | Fe solubility (N = 23) | Cu solubility (N = 21) |
|---|---|---|---|---|---|---|
| Mean (ng m$^{-3}$) | 478.9 | 61.8 | 20.2 | 13.8 | 14.6 % | 45.6 % |
| Median (ng m$^{-3}$) | 388.4 | 45.4 | 11.4 | 10.8 | 16.3 % | 44.0 % |
| Maximum (ng m$^{-3}$) | 1535.2 | 247.1 | 169.3 | 131.5 | 32.5 % | > 100 % |
| Minimum (ng m$^{-3}$) | 22.1 | 4.4 | LOD | LOD | 0.2 % | 3.3 % |
| RSD | 0.76 | 0.87 | 1.25 | 0.98 | 0.76 | 0.82 |

**Table 3:** Concentration of PM$_{2.5}$ particulates ( g m$^{-3}$) and transition metals (ng m$^{-3}$) in the ambient air of urban and suburban areas in different parts of the world.

|  | PM$_{2.5}$ mass | Total Fe | WS Fe | Total Cu | WS Cu |
|---|---|---|---|---|---|
| Present study, Atlanta (USA) | 10.4 | 203.7* | 20.2** | 30.7* | 13.8** |
| Previous study, Atlanta (USA)[a] |  |  | 21.1 |  | 13.8 |
| Patras (Greece)[b] | 17.4 | 124 | 11.9 | 7.28 | 2.67 |
| Megalopolis (Greece)[b] | 23.0 | 87 | 8.51 | 4.02 | 0.94 |
| Po Valley (Italy)[c] |  | 103 |  | 4.7 |  |
| Tampa (USA)[d] | 12.7 | 790 |  | 2.4 |  |
| Hong Kong (China)[e] | 29 | 160 |  | 15 |  |
| Agra (India)[f] | 104.9 | 1900 |  | 200 |  |

[a] Fang et al. (2015), [b] Manousakas et al. (2014), [c] Canepari et al. (2014), [d] Olson et al. (2008), [e] Hagler et al. (2007), [f] Kulshrestha et al. (2009)

* Based on ICP-MS measurement.

** Based on LWCC measurement



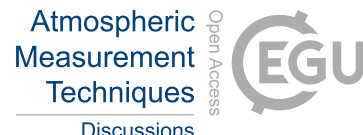


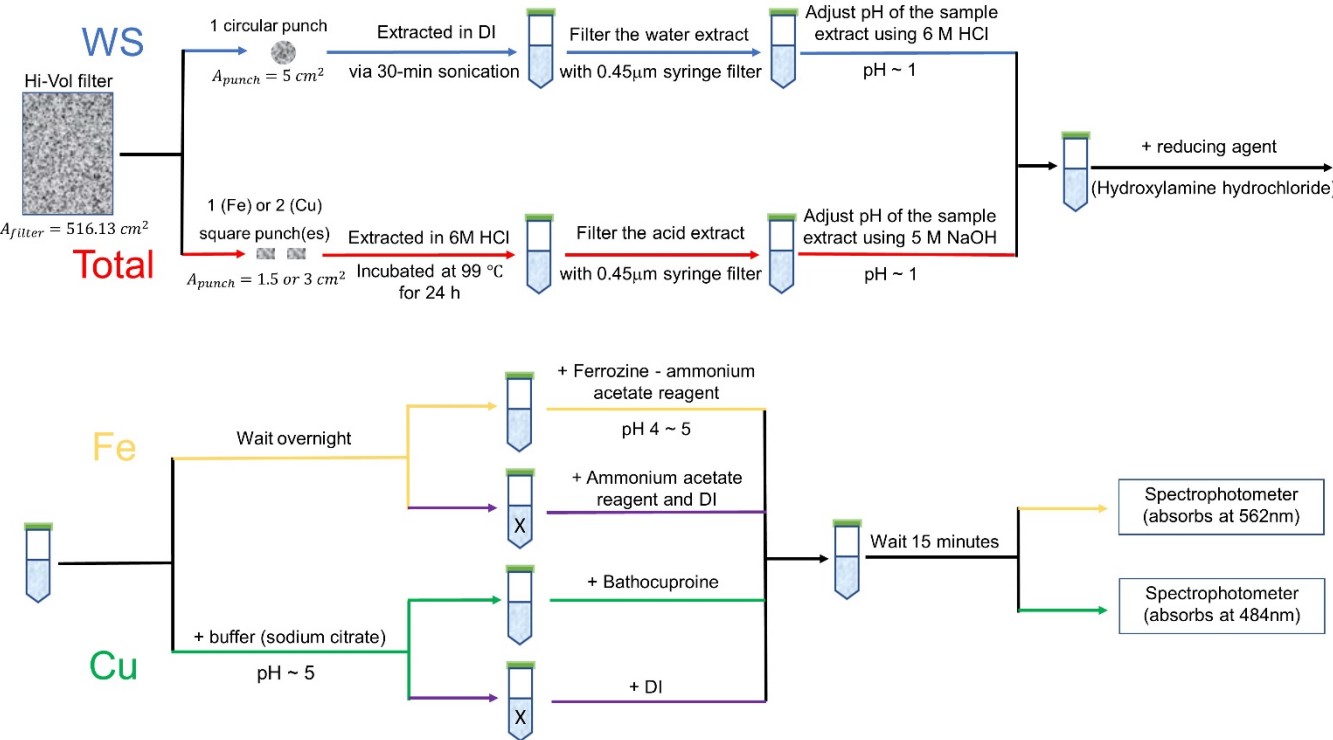

**Figure 1:** Schematic of the experimental procedures for the measurement of water-soluble (blue line) and total (red line) Fe
and Cu The upper schematic shows the processes common to both methods. The sample vial from this process is then analysed
for Fe or Cu via the process illustrated in the bottom schematic. In the bottom schematic the process for Fe (yellow line) or Cu
(green line) is given. The black line is common to both the WS and total measurement (in the upper schematic) or common to
both the Fe and Cu measurement (in the lower schematic). X refers to the liquid samples used to assess the interference of
other light-absorbing species at 562 or 484 nm (purple line).




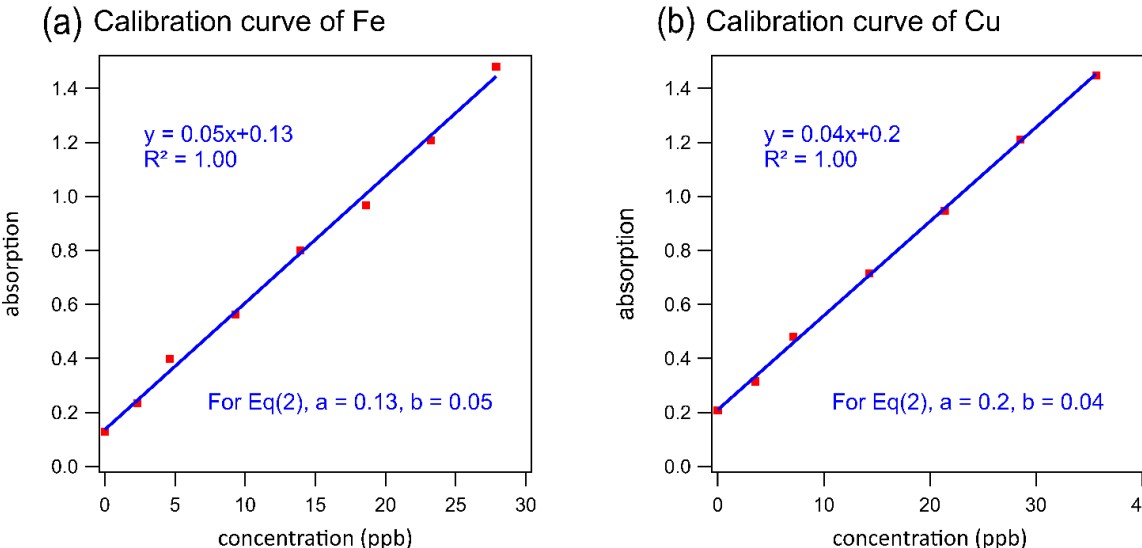

**Figure 2:** Calibration curves for (a) Fe(II) and (b) Cu(I) measurement by the LWCC system. Results of York regression are shown.





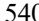

**Figure 3:** Comparison between LWCC method and ICP-MS method for (a) WS Fe, (b) total Fe, (c) WS Cu and (d) total Cu (results of York regression are shown, along with 1:1 ratio by black line). Bars represent the uncertainties of measurements.




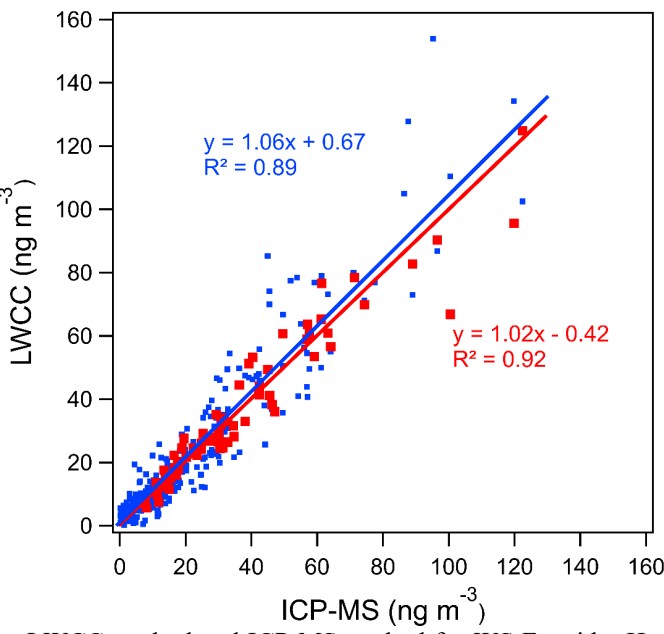

**Figure 4:** Comparison between LWCC method and ICP-MS method for WS Fe with pH adjustment of the light absorbing sample. The blue points and line represent the original method without ammonium acetate with pH of ~ 1. The red points and line refer to the improved method with ammonium acetate with pH 4 ~ 5 (results of York regression are shown).




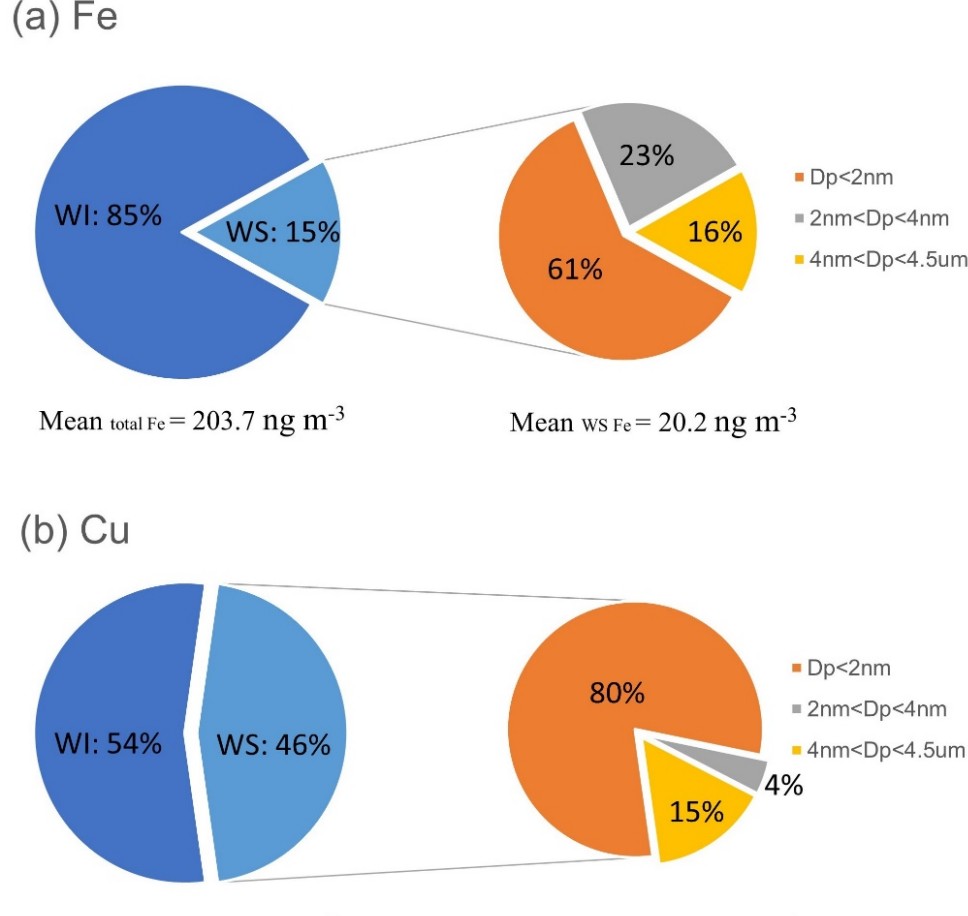

**Figure 5:** Components of PM$_{2.5}$ (a) Fe and (b) Cu. Water-soluble (WS) species are operationally defined by samples that pass
through a 0.45 µm pore syringe filter (shown in light blue). Water-insoluble (WI) are defined by the difference of all (total)
species of that metal and WS species (shown in dark blue). WS Fe (a) and WS Cu (b) concentrations were further separated in
ultrafiltration via 30,000 or 3,000 Dalton ultrafiltration filters (roughly corresponding to particle sizes of 4 and 2 nm,
respectively). Concentrations in different size ranges were determined by difference (statistics calculated on the differences).