# Peer review of "A Method for Liquid Spectrophotometric Measurement of Total and Water-Soluble Iron and Copper in Ambient Aerosols"

_Atmospheric Measurement Techniques, 2021_

## Author Comment (AC1)

We thank the reviewers for the insightful comments. We have addressed the reviewers' comments point by point as indicated below and revised the manuscript accordingly. The reviewers' comments are in italics and changes made to the manuscript are in quotation marks.

**Reviewer #1**

*General comments:*

*The authors described the relatively low-cost analytical method for measurement of water-soluble and total Fe and Cu in aerosols and analyzed the aerosol samples over Atlanta.*

*Such a high-frequency monitoring measurement is extremely useful for validation of the models and environmental assessment. I can recommend this paper for publication in Atmospheric Measurement Techniques and have minor comments to improve the paper.*

**Response:** We thank the reviewer for the positive and constructive comments. We appreciate the suggestion of improving the language. We have also addressed the specific comments by the reviewer accordingly.

*Specific comments:*

1) *Title: The reader might expect measurement of various chemical forms of iron and copper mentioned in introduction. Please consider rephrasing it by total and water-soluble, etc.*
   **Respon**se: We have changed the title, as suggested, to: A Method for Liquid Spectrophotometric Measurement of Total and Water-Soluble Iron and Copper in Ambient Aerosols

2) *p.7, l.191: Please discuss feasibility of in-situ measurements of ambient aerosols to investigate the speciation of WS Fe and WS Cu.*
   **Response:** We thank the reviewer for this comment. We have added "In-situ measurements of Fe(II) is feasible using this colorimetric method when connected to a PILS or similar system (Oakes et al., 2010; Rastogi et al., 2009). It is not feasible to perform near-real time measurements of Fe(III) due to its long reduction reaction time (overnight). Quantifying speciation of Cu is challenging due to the tendency of Cu(I) to undergo disproportionation in aqueous solutions (Johnson et al., 2015). In-situ measurements of Cu(I) is possible with the presence of reducing agents and masking ligand to inhibit Cu(II) interference (Moffett et al., 1985), but further studies are necessary to achieve an acceptable detection limit." to p.7, l.192-197.

3) *p.10, l.289: Please specify the chemical composition for the mass fraction of 5.63%.*
   **Response:** The abandance of iron element in the Earth's crust is 5.63%. To clarify, we rewrote the sentence in p.10, l.297 to "As expected the total Fe concentration was much higher than total Cu since Fe is a common element in the Earth's crust (mass fraction of iron element is 5.63 %) (Tomaszewski, 2017) and ubiquitous in mineral dust, while $PM_{2.5}$ Cu in urban environments is mainly derived from vehicle brake and tire wear, and tail-pipe emissions resulting from motor-oil impurities;".

4) *Conclusion: Please discuss feasibility of high-frequency monitoring of Fe and Cu in size-resolved aerosol and rainwater samples.*

   **Response:** We do not attempt to make high-frequncy monitoring in the size-resolved samples since the ultrafiltration procedure takes time (30-60 min) and cannot be done automatically.

5) *Data availability: How did you estimate the solubilities when the total and WS concentrations were blank?*

   **Response:** We thank the reviewer for the comment. We have added the following text for clarity "Those solubilities when total or WS concentrations were blank were estimated using half of the detection limit of the total or WS concentration." to p.10, l.292-293.

6) *Data availability: Please comment on the data such as 2017/3/20, which showed higher concentration through ultrafiltration than a 0.45 μm filter.*

   **Response:** We thank the reviewer for catching this. In most cases, the higher concentration through ultrafiltration than a 0.45 μm filter was because those two actual concentrations of that specific sample were similar and there were uncertainties during both measurements. We also deleted the data of the samples whose concentrations through 30k ultrafiltration were significantly higher than that through 0.45 μm filter (WS Fe of 3/20/2017 and 3/22/2017) and added the following text "The data of the samples whose concentrations through 30k ultrafiltration were significantly higher than that through 0.45 μm filter was removed." to p.10, l.306-307.

7) *Data availability: Please comment on the data such as 2017/4/10 and 2017/12/18, which showed higher solubilities than 100%.*

   **Response:** We thank the reviewer for pointing this out. The higher solubility than 100% was probably because the actual solubility was close to 100 % and there were uncertainties during both water-soluble and total measurements. To address the reviwer's concern, we changed higher solubilities than 100% to 1 and added "Due to uncertainties of both water-soluble and total measurements, a few calculated Cu solubilities were higher than 1 and were treated as 1 when calculating the average fraction of Cu." to p.10, l.290-292.

8) *p.16, l.510: Please correct the unit.*

   **Response:** Revised. Thanks!

9) *Supplement, p.2, l.32: Please correct "colloidal??".*

   **Response:** Revised. Thanks!

10) *Supplement, p.2, l.53: Please correct "Figure S8".*

**Response:** Revised. Thanks!

**References**

Johnson, D. K., Stevenson, M. J., Almadidy, Z. A., Jenkins, S. E., Wilcox, D. E., and Grossoehme, N. E.: Stabilization of Cu (I) for binding and calorimetric measurements in aqueous solution, Dalton Transactions, 44, 16494-16505, 2015.

Moffett, J. W., Zika, R. G., and Petasne, R. G.: Evaluation of bathocuproine for the spectrophotometric determination of the copper(I) in copper redox studies with applications in studies of natural waters, Analytical Chimica Acta., 175, 171-179, 1985.

Oakes, M., Rastogi, N., Majestic, B. J., Shafer, M., Schauer, J. J., Edgerton, E. S., and Weber, R. J.: Characterization of soluble iron in urban aerosols using near‐real time data, Journal of Geophysical Research: Atmospheres, 115, 2010.

Rastogi, N., Oakes, M. M., Schauer, J. J., Shafer, M. M., Majestic, B. J., and Weber, R. J.: New technique for online measurement of water-soluble Fe (II) in atmospheric aerosols, Environmental science & technology, 43, 2425-2430, 2009.

Tomaszewski, R.: ABUNDANCE OF ELEMENTS IN THE EARTH'S CRUST AND IN THE SEA, CRC Handbook of Chemistry and Physics and The Merck Index, Scientometrics, 112, 14-17, 2017.

---

## Author Comment (AC2)

**Reviewer #2**

**General comments:**

The manuscript needs a major revision. I think a high-throughput technique to measure Cu and Fe in PM2.5 will definitely contribute to the atmospheric research community. However, I do have a few concerns regarding the methodology part of this manuscript, for which I suggest a major revision before acceptance.

*PM2.5* is very heterogeneous in terms of elemental composition. You must consider the effect of other co-occuring elements when you are using a colorimetric method to measure a specific component (e.g. Cu or Fe). When multiple metal elements (such as Fe, Cu, Ni, Mn, Al and Co) are present at similarly high concentrations) simultaneously, is your colorimetric method still valid to measure Fe or Cu alone without being affected by other co-occuring elements? I saw you measured Mn, but your Mn concentration is very low, which would not interfere your results. But what is the threshold concentration of Mn and other transitional elements in air that makes your colorimeteic method invalid? In addition, it is commonly observed that *PM2.5* can contain a high level of both Cu and Fe. If we are interested in measuring both Cu and Fe, is your method able to separate Cu and Fe in concentration measurement? Does a high Cu or Fe concentartion affect the measurement of the other? What will be the detection limit when you have them cooccuring?

I did not see enough negative control and positive control groups. I suggest you include following controls, to make your work useful for the whole atmosphere research community.

1) Positive controls should be done to validate the colorimetric method for Cu and Fe. For example, if you have 100 ppb of Fe (II) with 100 ppb of WS Cu, 100 ppb WS Mn, 100 ppb WS Ni and 100 ppb WS Co, will your ferrozine exclude other co-occuring these other elements and accurately measure Fe (II)?

2) Similarly you need negative control groups as well. For example, if you have 0 ppb of Fe (II) with 100 ppb of WS Cu, 100 ppb WS Mn, 100 ppb WS Ni and 100 ppb WS Co, will your ferrozine method exclude other co-occuring these other elements and show the absence of Fe (II) ?

You need these control groups for ferrozine-Fe method and bathocuproine-Cu method. The discussion of artifacts due to effects of other elements is very important, because it informs other air researchers when your method is applicable.

**Response:** We thank the reviewer for the comments. We recognize the importance of the effect of other co-occuring elements when using a colorimetric method. We did a series of experiments following the reviewer's instruction. Since Co(II) and Cu (I) are the only metals other than iron which form colored species with ferrozine under the experiment condition (Stookey, 1970), we tested the interference of Cu, Co and Mn up to 100 ppb on the Fe measurement. Similarly, only Cu(II) has possible interference on Cu(I)-bathocuprione measurement in the presence of reducing agents (Moffett et al., 1985). In this study, we did not attempt to speciate Cu(I) and Cu(II). Therefore, we also tested the interference of up to 100 ppb Fe, Co and Mn on the Cu measurement. The tables below summarized the results of the control experiments (the concentrations in Table 1-3 represent the concentrations in the liquid):

Table 1: Interference of Fe, Mn and Co on Cu measurement.

|                 | 0 ppb Cu  | 30 ppb Cu |
|-----------------|-----------|-----------|
| 0 ppb mixture*  | 0.48 ppb  | 30.38 ppb |
| 20 ppb mixture  | -0.13 ppb | 30.55 ppb |
| 40 ppb mixture  | 0.16 ppb  | 30.77 ppb |
| 60 ppb mixture  | 0.00 ppb  | 30.72 ppb |
| 80 ppb mixture  | 0.53 ppb  | 30.92 ppb |
| 100 ppb mixture | -0.05 ppb | 29.95 ppb |

\* Mixed solution of equal amounts of Fe, Co and Mn in a solution originally containing 0 or 30 ppb Cu.

**Table 2:** Interference of Cu, Mn and Co on Fe measurement.

|                 | 0 ppb Fe  | 30 ppb Fe |
|-----------------|-----------|-----------|
| 0 ppb mixture*  | -0.12 ppb | 30.02 ppb |
| 20 ppb mixture  | 0.60 ppb  | 29.04 ppb |
| 40 ppb mixture  | 1.91 ppb  | 28.62 ppb |
| 60 ppb mixture  | 3.88 ppb  | 28.82 ppb |
| 80 ppb mixture  | 5.31 ppb  | 29.94 ppb |
| 100 ppb mixture | 6.89 ppb  | 32.00 ppb |
|                 |           |           |

\* Mixed solution of equal amounts of Cu, Co and Mn in a solution originally containing 0 or 30 ppb Fe.

According to Table 1, there's no interference of up to 100 ppb Fe, Co or Mn on Cu measurement considering the uncertainty of this method.

According to Table 2, there's possible interference of other metal ions on Fe measurement when Fe concentration is low. To investigate which metal ions cause this interference, we did further experiments and results are shown below:

**Table 3:** Measured Fe under different interference conditions.

| 0 ppb Fe |                                       |                                                                                                                                                         | 30 ppb Fe                                                                                                                                                                                                          |                                                                                                                                                                                                                                                                                            |                                                                                                                                                                                                                                                                                                                                                                                   |  |
|----------|---------------------------------------|---------------------------------------------------------------------------------------------------------------------------------------------------------|---------------------------------------------------------------------------------------------------------------------------------------------------------------------------------------------------------------------------|--------------------------------------------------------------------------------------------------------------------------------------------------------------------------------------------------------------------------------------------------------------------------------------------|-----------------------------------------------------------------------------------------------------------------------------------------------------------------------------------------------------------------------------------------------------------------------------------------------------------------------------------------------------------------------------------|--|
| Cu       | Co                                    | Mn                                                                                                                                                      | Cu                                                                                                                                                                                                                        | Co                                                                                                                                                                                                                                                                                         | Mn                                                                                                                                                                                                                                                                                                                                                                                |  |
| -0.79    | -0.90                                 | -0.12                                                                                                                                                   | 29.37                                                                                                                                                                                                                     | 30.79                                                                                                                                                                                                                                                                                      | 30.73                                                                                                                                                                                                                                                                                                                                                                             |  |
| -0.74    | -0.93                                 | 0.69                                                                                                                                                    | 29.85                                                                                                                                                                                                                     | 29.83                                                                                                                                                                                                                                                                                      | 30.12                                                                                                                                                                                                                                                                                                                                                                             |  |
| -0.31    | -0.94                                 | -0.02                                                                                                                                                   | 29.33                                                                                                                                                                                                                     | 28.81                                                                                                                                                                                                                                                                                      | 29.85                                                                                                                                                                                                                                                                                                                                                                             |  |
| 0.08     | -0.06                                 | -0.45                                                                                                                                                   | 29.93                                                                                                                                                                                                                     | 29.86                                                                                                                                                                                                                                                                                      | 30.65                                                                                                                                                                                                                                                                                                                                                                             |  |
|          | Cu
-0.79
-0.74
-0.31
0.08 | 0 ppb Fe           Cu         Co           -0.79         -0.90           -0.74         -0.93           -0.31         -0.94           0.08         -0.06 | 0 ppb Fe           Cu         Co         Mn           -0.79         -0.90         -0.12           -0.74         -0.93         0.69           -0.31         -0.94         -0.02           0.08         -0.06         -0.45 | 0 ppb Fe         Cu         Co         Mn         Cu           -0.79         -0.90         -0.12         29.37           -0.74         -0.93         0.69         29.85           -0.31         -0.94         -0.02         29.33           0.08         -0.06         -0.45         29.93 | 0 ppb Fe         30 ppb Fe           Cu         Co         Mn         Cu         Co           -0.79         -0.90         -0.12         29.37         30.79           -0.74         -0.93         0.69         29.85         29.83           -0.31         -0.94         -0.02         29.33         28.81           0.08         -0.06         -0.45         29.93         29.86 |  |

| 40 ppb  | 1.33 | -0.04 | -0.58 | 31.04 | 30.27 | 30.72 |
|---------|------|-------|-------|-------|-------|-------|
| 60 ppb  | 2.55 | 0.49  | -0.27 | 32.01 | 30.38 | 30.52 |
| 80 ppb  | 3.95 | 1.17  | 0.25  | 32.81 | 30.44 | 29.13 |
| 100 ppb | 6.01 | 1.62  | -0.05 | 33.34 | 31.17 | 29.61 |

According to Table 3, only high level Cu may have interference (> 2 ppb) on the Fe measurement. Since Cu concentration would be equal or less than Fe concentration in most ambient measurements, and the recommended Fe liquid concentration detected by the system are up to 40 ppb using this colorimetric measurement method, the interference of Cu would be less than 2 ppb, which could be neglected considering the uncertainty of this method.

To address the reviwer's concern, the sentences in p.9, 1.247-253 have been modified to "We also used a series of metal mixtures, which contained different levels of Fe, Cu, Co and Mn to test for possible interferences from the formation of colored complexes with other metals. The results from these experiments are summarized in Table S1-S3. For the Cu measurement, no interference from Fe, Co or Mn was observed with up to 100 ppb of mixture of Fe, Co and Mn present. Similarly, Co or Mn did not interfere with the Fe measurement with Co or Mn concentrations in the liquid up to 100 ppb. Cu had interference less than 2 ppb on the Fe measurement with Cu concentration in the liquid up to 40 ppb. In this study, the liquid concentrations of Cu from ambient samples were lower than 40 ppb, therefore, no interference of other metals on Cu or Fe measurement for the colorimetric methods.".

**References**

Moffett, J. W., Zika, R. G., and Petasne, R. G.: Evaluation of bathocuproine for the spectrophotometric determination of the copper(I) in copper redox studies with applications in studies of natural waters, Analytical Chimica Acta., 175, 171-179, 1985.

Stookey, L. L.: Ferrozin-a new spectrophotometric reagent for iron, Anal. Chem., 42, 779-781, 1970.

---

## Author Comment (AC3)

**Editor**

*General comments:*

*In the conclusion, detection limits of Fe and Cu in the air are provided. I think the detection limites in air depends on the volume of air sampled. Should the authors also provide dection limits in the solutions?*

**Response:** We thank the editor for the comment. We have added the following text for clarity "The detection limits in the solutions were 2.07 ppb for WS Fe, 2.05 ppb for WS Cu, 7.00 ppb for total Fe and 3.06 ppb for total Cu." to p.8, l 221-222.